# True amplification of spin waves in magnonic nano-waveguides

H. Merbouche [1], B. Divinskiy[1], D. Gouéré [2], R. Lebrun [2], A. El Kanj[2], V. Cros [2], P. Bortolotti[2], A. Anane [2], S. O. Demokritov[1] & V. E. Demidov [1] ✉

Magnonic nano-devices exploit magnons - quanta of spin waves - to transmit and process information within a single integrated platform that has the potential to outperform traditional semiconductor-based electronics. The main missing cornerstone of this information nanotechnology is an efficient scheme for the amplification of propagating spin waves. The recent discovery of spin-orbit torque provided an elegant mechanism for propagation losses compensation. While partial compensation of the spin-wave losses has been achieved, true amplification – the exponential increase in the spin-wave intensity during propagation – has so far remained elusive. Here we evidence the operating conditions to achieve unambiguous amplification using clocked nanoseconds-long spin-orbit torque pulses in magnonic nano-waveguides, where the effective magnetization has been engineered to be close to zero to suppress the detrimental magnon scattering. We achieve an exponential increase in the intensity of propagating spin waves up to 500% at a propagation distance of several micrometers.

Among novel promising nano-scale information technologies, the magnon-based information processing[1–3] occupies a special place due to the numerous advantages provided by magnetic excitations in solids – spin waves (SWs) and their quanta magnons. These excitations exist in the frequency interval from sub-GHz to sub-THz and possess wavelengths that can be as small as few tens of nanometers[4–6]. They can be guided and manipulated using simple submicrometer-wide strip magnetic waveguides and can be efficiently controlled by electric and magnetic fields, as well as by electric currents[7–12]. Thanks to these advantages, a large variety of novel magnonic devices and circuits for information processing[13–16] including neuromorphic and non-traditional computing systems[17,18] have been demonstrated in the recent years.

The main roadblock that is stalling magnonics from achieving large-scale integration of chips is the lack of a reliable scheme to perform signal restauration that allows magnonic logic gates to be cascaded. Indeed, since magnons have a finite lifetime, the total length of a magnonic device cannot be much larger than their attenuation length, which is expressed as the product of the characteristic magnon lifetime and the magnon group velocity. The most promising physical phenomenon that can help overcome this difficulty, is the spin-orbit torque in layered systems constituted by a magnetic film interfaced with a conductive material with strong spin-orbit coupling[19–21]. In such systems, the electric current injected in the conducting layer is converted into a transverse pure spin current, which flows into the magnetic film. The spin current then exerts a torque on the local magnetization that counteracts the natural damping torque[22], compensating, therefore, the losses and eventually amplifies the magnetic excitations[23].

Although this approach seems straightforward, in practice it only works well in the limit of low currents, below the critical current, at which the complete damping compensation takes place. Such partial damping compensation can be easily achieved and was reported for standing and propagating spin waves[24–33]. In the case of propagating spin waves, it was shown to enable partial compensation of the spatial attenuation, which increases the spin-wave amplitude at a given distance from the source. However, the true amplification of spin waves by spin currents, which manifests itself by an

[1]Institute of Applied Physics, University of Muenster, Corrensstrasse 2-4, 48149 Muenster, Germany. [2]Laboratoire Albert Fert, CNRS, Thales, Université Paris-Saclay, 91767 Palaiseau, France. ✉e-mail: demidov@uni-muenster.de

exponential increase in the amplitude of spin waves with distance, has not yet been achieved.

Magnon-magnon interaction is the main reason why the true spin-torque amplification of SWs is difficult to achieve. In standard magnetic systems, a spin-torque induced damping compensation results in the amplification of all magnon modes, which leads to the excitation of large-amplitude coherent and/or incoherent magnetization auto-oscillations[28], which causes additional nonlinear scattering of the signal spin wave. This is the reason why in previous experiments the signal spin wave was not truly amplified, although the natural magnetic damping was compensated by the spin-torque caused by spin current[26,33].

In this work, we show that the fundamental limitations preventing the amplification of spin waves by spin currents can be overcome using a combination of nonlinearity control and temporal separation. Using microscopy imaging of propagating spin waves, we demonstrate that this approach allows one to achieve an exponential spatial increase of the amplitude of a spin wave as it propagates in a nano-fabricated waveguide based on a magnetic insulator with perpendicular anisotropy. We show that by tuning the angle of the static magnetic field, it is possible to minimize the detrimental nonlinear scattering of the signal spin wave from spin-current induced auto-oscillations. We also show that the complete elimination of the scattering, necessary to achieve the amplification, requires synchronizing the spin-current pulse with the spin wave. Otherwise said, the pulse of the signal spin wave must propagate its way before the spin current drives the spin system into a strongly excited nonlinear state. Our findings open new horizons for the development of efficient cascaded integrated magnonic information-processing circuits supported by direct on-chip spin-wave amplification.

## Results

### Studied system and approach

Figure 1a and b show the schematics of the experiment. We study a 500-nm wide magnonic waveguide patterned by electron-beam lithography and ion etching from a bilayer constituted by a 20-nm thick film of Bi-doped Yttrium Iron Garnet (BiYIG) $(Bi_1Y_2Fe_5O_{12})$[34] and a 6-nm thick Pt film. The waveguide is magnetized by a static magnetic field $H_0$. The field is applied perpendicular to the axis of the waveguide at an angle $\theta$ relative to its plane. The BiYIG film exhibits a strain-induced perpendicular magnetic anisotropy (PMA) with the effective anisotropy field $H_a = 1.8$ kOe, which is close to the saturation magnetization of the film $4\pi M_s = 1.5$ kG. This leads to a significant reduction of the ellipticity of magnetization dynamics and thus to a minimization of detrimental nonlinear spin-wave interactions[35]. The spin waves are inductively excited using a 300-nm wide and 80-nm thick Au antenna oriented perpendicular to the waveguide. The antenna is electrically isolated from the Pt layer by a 50 nm thick layer of $SiO_2$. The excitation microwave current with a carrier frequency $f$ and a power of 0.1 mW, which is sufficiently small to ensure a linear excitation regime, is applied in the form of 100-ns wide pulses with the repetition period of 5 μs. Simultaneously, we apply 200-500-ns long pulses of dc current $I$ through the Pt layer on top of the magnetic waveguide. Due to the spin-Hall effect[36,37] in Pt, electrons with opposite orientations of the magnetic moment scatter toward opposite surfaces of the Pt film (see inset in Fig. 1a). As a result, the in-plane dc current in Pt is converted into an out-of-plane pure spin current $I_s$, which is injected into the BiYIG film. This injection results in a compensation of the natural magnetic damping in the magnetic material[22] and is expected to enable amplification of the propagating spin waves at sufficiently large $I$. The propagation of spin-wave pulses is visualized with high spatial and temporal resolution using the micro-focus Brillouin light scattering (BLS) spectroscopy[10] (see Methods for details). This technique yields a signal referred to as BLS intensity, which is proportional to the intensity of spin waves at the position where the probing laser light is focused. By synchronizing the detection of the scattered probing light with the excitation pulses, we obtain the possibility to directly analyze the propagation of spin-wave pulses in the space and time domain.

### Evidence of spin-wave amplification

Figure 1c, d demonstrate the main result of our work – the true amplification of spin-wave pulses by the spin current, which is

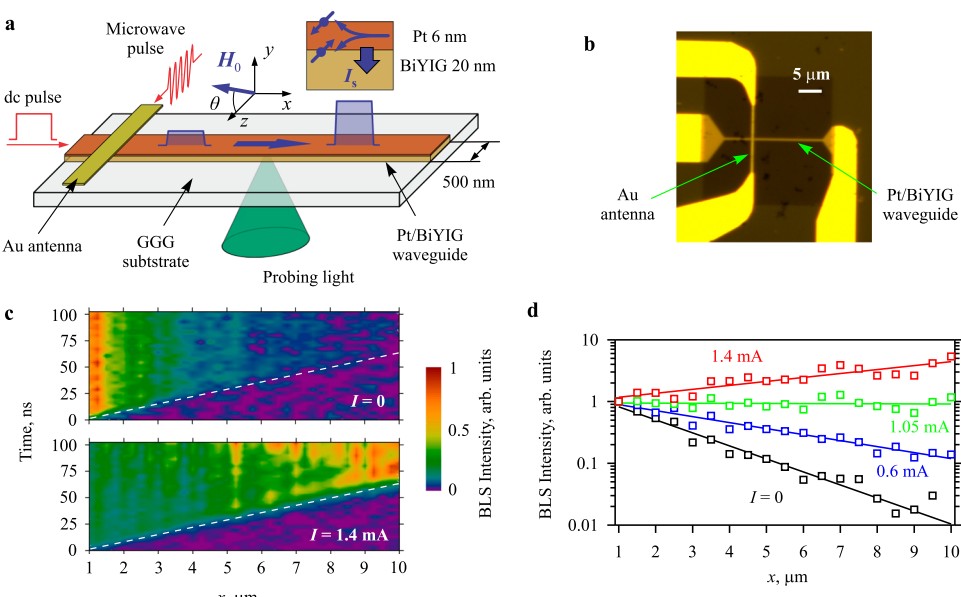

**Fig. 1 | Implementation of spin-wave amplification. a** Schematics of the experiment. Spin-wave pulses are excited by a Au antenna and propagate in a 500-nm wide waveguide fabricated from a BiYIG(20 nm)/Pt(6 nm) bilayer. In-plane dc current flowing in the Pt layer is converted into an out-of-plane pure spin current $I_s$ (inset), which is injected into the BiYIG film and exerts anti-damping torque on the magnetization. **b** Optical micrograph of the sample. **c** Normalized BLS maps of the spin-wave intensity in the space-time coordinates recorded at $I = 0$ and 1.4 mA, as labeled. Dashed lines show the spatio-temporal shift of the edge of the spin-wave pulse corresponding to the group velocity of 135 m s$^{-1}$. **d** Spatial dependence of the intensity of the spin-wave pulse measured at different dc currents, as labeled. Symbols show the experimental data. Solid straight lines show the exponential fit. The data are obtained at $f = 5.025$ GHz and $H_0 = 1.8$ kOe applied at $\theta = 30°$.

achieved at $H_0 = 1.8$ kOe applied at $\theta = 30°$. As will be discussed below, the choice of the angle $\theta = 30°$ is crucial to achieve the desired effect, since nonlinear frequency shift and residual ellipticity are minimized under these conditions, resulting in the strong suppression of detrimental nonlinear interactions. Figure 1c shows the color-coded spin-wave intensity in the space-time coordinates for $I = 0$ and 1.4 mA. The origin of the time axis is aligned with the beginning of the leading edge of the microwave pulse, while its span is equal to the pulse width of 100 ns. The origin of the space axis is aligned with the center of the antenna. The data obtained at $I = 0$ demonstrate a damped propagation of the spin-wave pulse unaffected by the spin current. The leading edge of the pulse linearly shifts in time (dashed line), which corresponds to the constant group velocity of 135 m s⁻¹, while the intensity of the pulse strongly reduces between $x = 1$ and $10\,\mu m$ due to the natural magnetic damping. In contrast, the data obtained at $I = 1.4$ mA show a significant increase in the intensity of the pulse during propagation. We note that the group velocity remains unchanged in this regime. This indicates that the application of the dc current does not modify other parameters of the spin wave, except for damping, which obviously becomes negative.

Figure 1d illustrates the spatial attenuation/amplification of spin waves at different $I$. As seen from these data, at all currents, the spatial dependence of the spin-wave intensity is exponential (note the logarithmic scale of the vertical axis) and can be described by $e^{\kappa x}$, where $\kappa$ is the intensity decay constant. At $I = 0$, the intensity of the spin wave reduces by about two orders of magnitude at the propagation distance $x = 1$–$10\,\mu m$ ($\kappa \approx 0.5\,\mu m^{-1}$). Taking into account the measured group velocity $v_g = 135$ m s⁻¹ and the frequency of spin waves $f \approx 5.0$ GHz, this corresponds to the Gilbert damping parameter $\alpha = \kappa v_g/(4\pi f) \approx 1.2 \times 10^{-3}$, which is typical for BiYIG/Pt bilayers[38]. With the increase in $I$, the spatial attenuation becomes weaker. The attenuation completely vanishes at $I = 1.05$ mA, resulting in a decay-free propagation regime. Further increase in $I$ leads to an exponential increase of the intensity of spin waves in space. At the maximum current used in the experiment ($I = 1.4$ mA), the intensity increases by an almost a factor of 5 over the propagation distance $x = 1$–$10\,\mu m$. Note that in our experiments, the current was limited to 1.4 mA to ensure that the sample does not degrade during long measurements. This maximum current corresponds to a current density of only $4 \times 10^{11}$ A m⁻². A significantly higher current could therefore be considered for future optimized devices (see, e.g., ref. 39, where the current densities of more than $10^{12}$ A m⁻² were achieved in BiYIG/Pt structures).

## Effect of the angle of the static magnetic field

Although the results discussed above might lead to the conclusion that the amplification by spin current is straightforward to implement, a detailed consideration shows that this is not a trivial task and is possible only within a certain parameter window. Figure 2a shows the current dependence of the decay constant obtained for two orientations of the static magnetic field $\theta = 0$ and $30°$. At $\theta = 30°$, the dependence is linear over the entire range $I = 0$–1.4 mA, as expected for the effects of the spin transfer torque on the magnetic damping[40]. It crosses zero at $I_C = 1.07$ mA, which marks the transition to the amplification regime. In contrast, at $\theta = 0$ the dependence is linear only at $I < I_C$. At larger currents, the decay constant saturates, and its value remains negative, indicating that the amplification regime is never achieved. Note that, while changing in the experiment $\theta$ from 0 to $30°$, we simultaneously adjust the magnitude of $H_0$ from 2.0 to 1.8 kOe in order to maintain a constant frequency of the ferromagnetic resonance $f_{FMR} \approx 4.99$ GHz. This ensures that the experimental conditions are not strongly altered by the change in $\theta$.

We emphasize that previous studies of microscopic YIG/Pt waveguides without PMA also showed the absence of the amplification at currents exceeding the threshold for complete damping

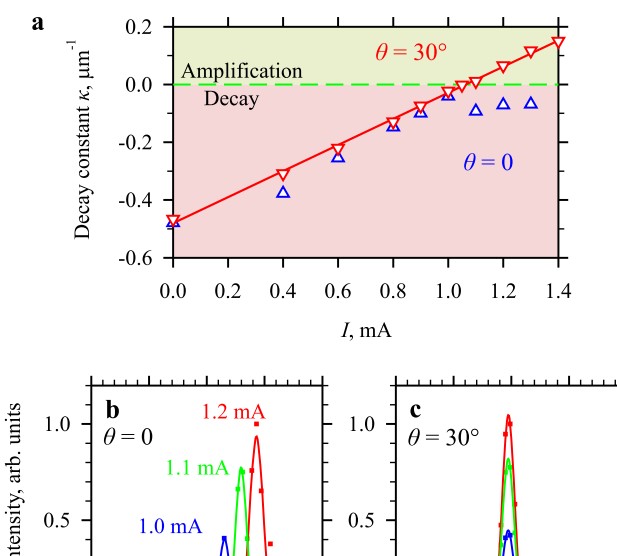

**Fig. 2 | Effect of the angle of the static magnetic field on amplification and auto-oscillations. a** Current dependence of the decay constant of spin-wave intensity obtained at $\theta = 0$ and $30°$, as labeled. Symbols show experimental data. Solid line is the linear fit of the data at $\theta = 30°$. **b, c** BLS spectra of magnetization auto-oscillations recorded at the labeled values of the dc current at $\theta = 0$ and $30°$, respectively, without applying microwave pulses to the antenna. The data for $\theta = 0$ and $30°$ are obtained at $H_0 = 2.0$ and 1.8 kOe, respectively, to compensate for the frequency shift of the spin-wave dispersion spectrum.

compensation[26,33]. As mentioned above, this was associated with the onset of intense magnetic auto-oscillations under conditions of complete damping compensation. Although these auto-oscillations have a frequency different from the frequency of spin waves to be amplified, the nonlinear scattering of these waves from intense auto-oscillations prevents their amplification. Similarly, we also observe an onset of auto-oscillations in the BiYIG/Pt waveguide both at $\theta = 0$ and $30°$, as shown in Fig. 2b and c, in which we observe an appearance of a narrow intense spectral peak at $I > 1$ mA indicating the excitation of auto-oscillations induced by the spin current. We use the observed signals to characterize the nonlinear properties of our system at different angles $\theta$. The main difference between the two cases shown in Fig. 2b and c lies in the nonlinear frequency shift of the auto-oscillation peak with increase of the auto-oscillation amplitude. While at $\theta = 0$, the auto-oscillation frequency noticeably increases with $I$, at $\theta = 30°$ it remains almost constant. These behaviors are caused by the interplay between the effects of PMA and the effects of the dipolar demagnetizing fields on the spectrum of magnetic excitations in the BiYIG film (see Supplementary Note 1 for details). We emphasize that the vanishing nonlinear frequency shift is expected to be accompanied by a decrease in the ellipticity of the magnetization precession. This latter parameter is indeed known to be one of the most important factors for nonlinear coupling of spin waves responsible for their scattering[35].

Nonlinear scattering is the exchange of energy between different spin waves caused by their nonlinear interaction. In the linear approximation, all magnon (spin-wave) eigenstates are orthogonal and independent of each other. However, with an increase in the amplitude of magnetization precession, nonlinear coupling mechanisms become active. The most efficient coupling mechanism is associated with the parametric interaction of spin waves, which requires the precession to be elliptical. As illustrated in Fig. 3a, the large-amplitude elliptical

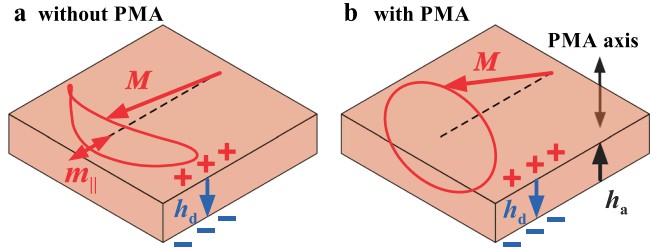

**a without PMA**

**b with PMA**

PMA axis

**Fig. 3 | Effects of PMA on the ellipticity of magnetization precession. a** In magnetic films, the magnetization precession is strongly elliptical due to the dynamic dipolar demagnetizing field $h_d$. **b** In films with PMA, the dipolar field $h_d$ can be compensated by the effective field of the anisotropy $h_a$ resulting in a decrease in the ellipticity.

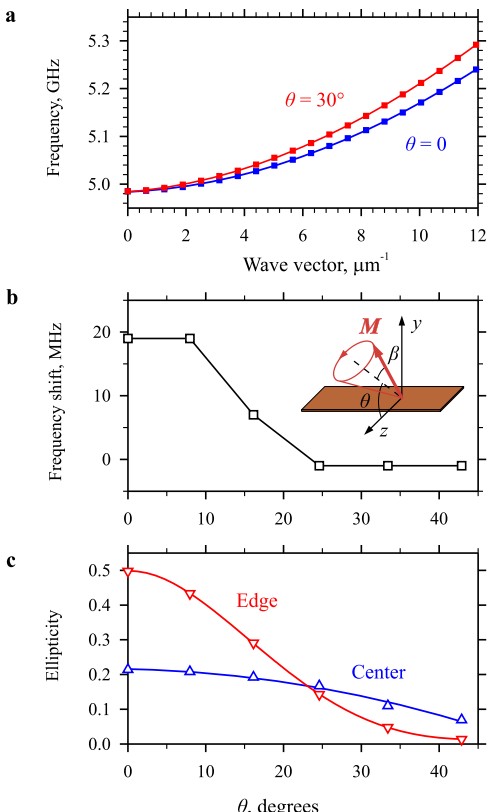

**Fig. 4 | Results of micromagnetic simulations of spin-wave dynamics in the nano-waveguide. a** Dispersion curves of spin waves calculated at $\beta = 0.1°$ and $\theta = 0$ and 30°, as labeled. **b** Angular dependence of the nonlinear frequency shift of the spin-wave spectrum calculated as the difference in the frequency of spin waves for the magnetization precession cone angles $\beta$ of 10° and 0.1°. **c** Angular dependences of the ellipticity of the magnetization precession in the center and at the edge of the nano-waveguide. Curves are guides for the eye. These simulations points toward $\theta = 30°$ as a field angle, at which both the nonlinear frequency shift and the ellipticity almost vanish.

precession of the magnetization vector leads to the appearance of a component of the dynamic magnetization $m_\parallel$, parallel to the precession axis. This component represents a periodic modulation of the parameter of the system – static magnetization. If the modulation amplitude is large enough, the modulation can lead to parametric excitation of pairs of spin waves whose frequencies and wave vectors are different from those of the initial spin wave. In other words, different spin waves become coupled.

In magnetic films, the ellipticity of the magnetization precession is due to the dynamic dipolar demagnetizing field $h_d$, which is antiparallel

to the out-of-plane component of the dynamic magnetization (Fig. 3a). In films with PMA (Fig. 3b), the dipolar field $h_d$ can be compensated by the effective field of the anisotropy $h_a$, which is oriented parallel to the out-of-plane component of the magnetization. This results in a decrease in the ellipticity. In the case of a narrow waveguide, the trajectory of the magnetization vector is additionally affected by in-plane demagnetizing fields caused by the lateral edges. By varying the angle of the static magnetic field $\theta$, one can control the relative contributions of all these fields and minimize ellipticity and, as a consequence, nonlinear scattering.

## Micromagnetic simulations

In order to get better insight into the observed behaviors, we perform micromagnetic simulations using the MuMax3 software[41] (see Methods for details). First, we calculate the dispersion spectrum of spin waves in the waveguide for various angles $\theta$ in a small-amplitude regime corresponding to the magnetization precession cone angle $\beta = 0.1°$ (Fig. 4a). Similar to the experiment, we vary the magnitude of $H_0$ from 2.0 ($\theta = 0$) to 1.8 kOe ($\theta = 30°$) to maintain a constant frequency of the ferromagnetic resonance. As seen From Fig. 4a, within the entire range $\theta = 0 - 30°$, the dispersion curves show an increase in the frequency with increasing wave vector. The increase in the angle $\theta$ only leads to a slight increase in the slope of the curves (increase in the group velocity). Second, we calculate the dispersion curves for $\beta$ increasing from 0.1 to 10° and determine the nonlinear frequency shift. Figure 4b shows the nonlinear frequency shift calculated as the difference in the frequency of spin waves with zero wave vector obtained at $\beta = 10°$ and 0.1°, as a function of the angle of the static magnetic field $\theta$. These results are in good agreement with the experimental data (Fig. 2b, c). Both experiment and simulations show that the frequency shift is positive at $\theta = 0$ and almost vanishes with the increase of $\theta$ to 30°. Note, that the absolute value of the nonlinear shift observed experimentally at $\theta = 0$ (Fig. 2b) is about 200 MHz, which is significantly larger than the values obtained from simulations. This is not surprising, since, in the experiment, the frequency shift is determined not only by an increase in the precession angle, but also by an increase in the temperature of the sample due to heating by electrical current (see Methods for details).

Additionally, the simulations allow us to obtain information about the ellipticity of the magnetization precession, which is hardly accessible in the experiment. We calculate the ellipticity using the standard expression $\varepsilon = 1 - \frac{|m_{min}|^2}{|m_{max}|^2}$, where $m_{min}$ and $m_{max}$ are the smallest and the largest values of the dynamic magnetization over the precession cycle. Note that due to the complex distribution of the dynamic demagnetizing field across the width of the nano-patterned waveguide, the ellipticity differs significantly at the center and at the edges of the waveguide. As seen from Fig. 4c, it also shows essentially different angular dependences. At the center of the waveguide, the ellipticity is relatively small at $\theta = 0$ and slowly reduces with the increase in the angle. In contrast, at the edge of the waveguide, the ellipticity is as large as 0.5 at $\theta = 0$. This large ellipticity is generally expected to result in a strong nonlinear spin-wave coupling and is likely the reason for the nonlinear suppression of the effects of the spin current observed in the experiment at $\theta = 0$. As seen from Fig. 4c, the edge ellipticity quickly decreases with the increase in $\theta$, which can explain the absence of the adverse influence of intense auto-oscillations on the spin-current amplification observed in the experiment at $\theta = 30°$.

The data of simulations clearly show that shape effects in nano-patterned waveguides strongly influence the processes of amplification of propagating spin waves by pure spin currents by enhancing the nonlinear spin-wave scattering. In an extended magnetic film, the scattering can be efficiently suppressed by tuning the strength of PMA, which allows one to exactly compensate the dipolar

demagnetizing fields and to obtain almost circular magnetization precession trajectory[35,42] (see Fig. 3). On the contrary, in the case of a patterned waveguide, one additionally needs to tune the angle of the static field to compensate for the shape effects. However, even this approach cannot provide a complete suppression of ellipticity over the entire cross section of the waveguide. As a result, nonlinear-shift management strategies alone cannot fully eliminate the detrimental nonlinear scattering of the propagating spin wave.

## Temporal management

We develop an active strategy to achieve a full suppression of non-linear scattering process and true spin-wave amplification based on a time synchronization between the spin-wave signal and the spin-current pulse. In particular, due to residual interaction of the signal spin wave with excited auto-oscillations, amplification can be achieved only within a certain time interval after the spin current is applied. Figure 5a shows representative temporal dependences of the intensity of magnetic oscillations after the start of the spin-current pulse. The dependence recorded at $I = 1.4$ mA corresponds to the case when the damping is completely compensated resulting in the development of auto-oscillations, while the dependence recorded at $I = 1.0$ mA corresponds to the case of incomplete compensation. Both dependences exhibit a rapid intensity increase (note the logarithmic scale of the vertical axis) directly after the start of the dc pulse. This increase reflects the known effect of the enhancement of magnetic fluctuations by the spin current[43] and does not indicate an onset of auto-oscillations. At $I = 1.4$ mA, the initial increase is followed by an exponential growth of the intensity (dashed line in Fig. 5a). The growth rate linearly depends on $I$ (Fig. 5b), as expected for auto-oscillations induced by the spin-transfer torque[40]. Note that the linear dependence in Fig. 5b crosses zero at $I = 1.07$ mA, which is consistent with the critical current necessary to observe the amplification of propagating spin waves (Fig. 2a).

Similarly to any system exhibiting auto-oscillations, the exponential growth of auto-oscillations is followed by saturation caused by an increase in the nonlinear scattering (nonlinear damping) at large oscillation amplitudes[40]. To avoid this nonlinear scattering process, the spin-wave amplification should happen before the saturation takes place, as demonstrated by the data in Fig. 5c. The figure shows the spatial dependences of the intensity of the signal spin wave obtained from measurements performed with spin-wave pulses propagating either during the transient-regime interval or in the saturation steady-state regime. As seen from these data, we observe the true spin-wave amplification only in the transient regime and an attenuation in the saturation regime. This key result evidences that, even in systems with minimized nonlinear scattering, efficient amplification is possible only with spin-current pulses that are short enough to not drive the spin system into the strongly nonlinear saturated state.

## Frequency dependence

Now, we discuss the robustness of the amplification regime by changing the frequency and the group velocity of spin waves. We vary the frequency of the microwave signal applied to the antenna and determine the group velocity $v_g$ of the spin-wave pulses from the spatio-temporal shift of the edge of the spin-wave pulse (Fig. 6a). As seen from these data, the group velocity increases by about a factor of two in the frequency range 5.0–5.2 GHz. In Fig. 6b, we then show the frequency dependences of the spatial spin-wave decay constant obtained in the attenuation regime (point-down triangles) and in the amplification regime (point-up triangles). The twofold decrease in the decay constant observed in the attenuation regime, is a natural consequence of the twofold increase in the group velocity. A less obvious result is the observed decrease in the magnitude of the decay constant in the amplification regime, indicating less efficient amplification of faster

waves. However, this result can be reproduced using a simple theoretical model. We calculate the spatial decay constant as

$$\kappa = \Gamma_r / 2v_g, \tag{1}$$

where $\Gamma_r$ is the temporal relaxation rate of the magnetization precession, which is determined from the data in Fig. 5b. The obtained dependencies (solid curves in Fig. 6b) agree well with the experimental data.

## Coherence of the amplification process

We now prove that the amplification process does not disturb the coherence of the amplified spin wave. For this, we perform phase-resolved BLS experiments. We measure the interference of light scattered from the spin wave with reference light modulated by the microwave signal used to excite spin waves with the phase shifted by $\Delta\varphi$. We vary the phase shift $\Delta\varphi$ from 0 to 360° and obtain a signal proportional to $A_{sw} \cos(\Delta\varphi)$, where $A_{sw}$ is the amplitude of the spin wave phase-locked (coherent) to the signal applied to the excitation antenna. Figure 7 shows the interference curves recorded at the maximum current $I = 1.4$ mA at the distance $x = 1\,\mu$m and $x = 10\,\mu$m. As seen from these data, the amplitude of the interference oscillations, which is proportional to the amplitude of the coherent spin wave, increases by a factor of 2.35. This increase in the amplitude agrees well with the fivefold increase in the intensity (intensity is equal to the square of the amplitude) observed in Fig. 1d. In other words, the observed increase in the intensity is associated with the amplification of the coherent signal and is not due to an increase in the incoherent background. This clearly indicates that the demonstrated intensity amplification fully preserves the coherence of the spin wave.

In conclusion, our results provide direct experimental evidence for the possibility of the true spatial amplification of propagating spin waves by spin currents in magnonic nano-waveguides. They unambiguously identify the physical phenomena that prevented amplification in previous experimental studies and show how these detrimental effects can be overcome in practice. These findings open new avenues for the field of nano-magnonics by demonstrating a simple and energy-efficient approach for the on-chip amplification of propagating spin waves, which can be used in most of nanoscale magnonic devices. The possibility to directly amplify propagating spin waves with a small dc current enables the implementation of complex cascadable magnonic nano-circuits with a large fan-out that do not require the energy-consuming conversion of spin waves into radio-frequency electronic signals for compensation of propagation losses, which is expected to significantly advance the practical realization of magnon-based computing platforms.

## Methods

### Sample fabrication

The BiYIG film was grown by pulsed laser deposition (PLD) using stoichiometric target BiYIG on (111) substituted Gallium Gadolinium Garnet (sGGG) substrate with a lattice parameter of 1.2497 nm. The distance between the target and the substrate was 44 mm. The deposition was performed using a frequency tripled Nd:YAG laser ($\lambda = 355$ nm) with a 2.5-Hz repetition rate and a fluency of about 1 J cm$^{-2}$. The uniaxial anisotropy is set by the choice of the substrate temperature[42] (420 °C). Prior to the deposition, the substrate was annealed at 700 °C under 0.4 mbar of $O_2$. The growth was performed at 0.25 mbar $O_2$ pressure. Finally, the sample was cooled down under 300 mbar of $O_2$. No post annealing was performed. The Pt layer was deposited using dc magnetron sputtering. Prior to Pt deposition, the BiYIG film was slightly etched with $O_2$ to promote surface spin-transparency. Detailed material characterizations can be found in ref. 42 and nanofabrication procedure in ref. 44. After waveguide nanofabrication, a 50 nm thick $SiO_2$

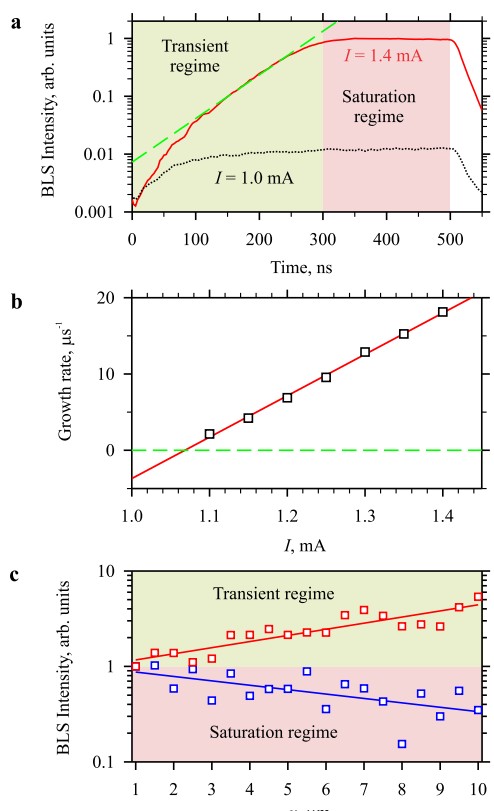

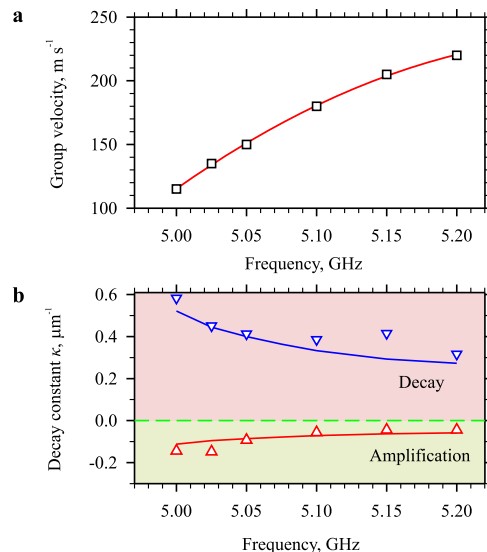

**Fig. 6 | Effects of the velocity of spin waves on the amplification efficiency.**
**a** Frequency dependence of the group velocity. Symbols show the experimental data. Curve is the guide for the eye. **b** Frequency dependences of the decay constant obtained at $I = 0$ (point-down triangles) and at the maximum current (point-up triangles). Note that negative decay constants correspond to the spatial amplification of the wave. Curves show the results of calculations using Eq. (1). The data are obtained at $H_0 = 1.8$ kOe applied at $\theta = 30°$. The largest amplification efficiency is obtained for the slowest spin-waves.

**Fig. 5 | Time constraints of the amplification process. a** Temporal dependences of the intensity of auto-oscillations after the start of the dc pulse recorded at $I = 1.4$ mA (solid curve). The dependence shows the transient regime and the saturation regime with a cross over at about 300 ns. Dashed curve shows the exponential increase of the intensity of auto-oscillations. Temporal dependence of the intensity of magnetic fluctuations (dotted curve) recorded at $I = 1.0$ mA is shown for reference. **b** Current dependence of the exponential growth rate of current-induced auto-oscillations. Symbols show the experimental data. Line is a linear fit. **c** Spatial dependences of the intensity of the signal spin-wave for the cases when the spin-wave pulse is applied during the time interval corresponding to the transient and saturation regime, as labeled. Symbols show the experimental data. Solid lines show the exponential fit. The data are obtained at $f = 5.025$ GHz and $H_0 = 1.8$ kOe applied at $\theta = 30°$.

layer (dark square in Fig. 1b) was deposited at room temperature using Ion Beam assisted deposition. Subsequently, a Ti(10 nm)/ Au(120 nm) layer was deposited and structured using electron beam lithography and lift-off.

## Micro-focus BLS measurements
The measurements were performed at room temperature. The detection of propagating spin waves and current-induced auto-oscillations was performed using the analysis of the inelastic scattering of laser light from magnetic excitations. The probing laser light had the wavelength of 532 nm and the power of 0.1 mW. The light was focused through the transparent sGGG substrate onto the BiYIG film using a high-performance corrected microscope objective lens with the magnification of 100 and a numerical aperture of 0.85. The light scattered from magnetic excitations was collected by the same lens and sent for analysis to a six-pass Fabry-Perot interferometer. The intensity of the scattered light (BLS intensity) was proportional to the intensity of magnetization oscillations at the position of the probing spot. The temporal resolution was achieved by synchronizing the detection of the scattered light with the excitation of spin waves. By moving the probing laser spot along the waveguide, detection of the

spin-wave intensity with simultaneous spatial and temporal resolution was implemented. To achieve high spatial accuracy (<50 nm), active stabilization of the sample position was used.

## Heating by the electrical current
The heating of the sample by the electrical current in Pt can be estimated based on the measurements of its electrical resistance on the current strength (Supplementary Fig. 1). Assuming the temperature coefficient of the electrical resistance in thin Pt films of $7 \times 10^{-4}$ K$^{-1}$ (ref. 45), the temperature increase in the continuous-current regime is 130 K at $I = 1.4$ mA. To reduce the effects of the heating, all the measurements were performed in the pulsed regime with an on/off ratio ≥10 and the pulse duration <500 ns.

## Micromagnetic simulations
We numerically simulated spin-wave dynamics in a 500 nm wide and 20-nm thick strip waveguide with the length $L = 10$ μm. The computation domain was discretized into 10 nm × 10 nm × 10 nm cells with periodic boundary conditions at the ends of the waveguide. The cell size was chosen to be smaller than the exchange length in YIG, which can be estimated as ≈18 nm. The magnetization dynamics was excited by spatially-periodic deflection of magnetic moments from their equilibrium orientation in the direction parallel to the waveguide axis. The spatial period of the deflection defined the wave vector of the excited spin waves $k_n = 2\pi n/L$ ($n = 0,1-25$), and the deflection angle defined the angle (amplitude) of the magnetization precession. We simulated the free dynamics of the magnetization caused by the initial deflection over a time interval of 1 μs with a fixed temporal step of 1 ps (saving step – 50 ps). From the Fourier analysis of the temporal dependences of the $x$-component of magnetization, we determined the frequency corresponding to a given wave vector and the amplitude of spin waves. This approach allowed us to calculate the amplitude-dependent dispersion spectrum of spin waves and its nonlinear shift caused by an increase in the precession angle.

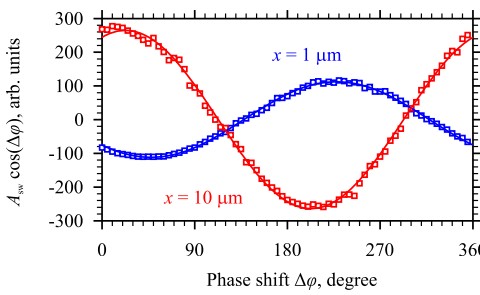

**Fig. 7 | Evidence of the coherence of the amplification process.** The dependences show the amplitude of the interference of the BLS signal carrying information about the spin wave with the reference light modulated by the signal used for the excitation of the spin wave with the phase shifted by $\Delta\varphi = 0 - 360°$. The data are obtained at $x = 1\ \mu m$ and $x = 10\ \mu m$, as labeled, at $I = 1.4$ mA and $f = 5.025$ GHz. Symbols show the experimental data. Curves show the fit by a sinusoidal function.

## Data availability

The data that support the findings of this study are available from the corresponding author upon reasonable request.

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

## Acknowledgements
This work was supported by in part by the Deutsche Forschungsgemeinschaft (DFG, German Research Foundation) – project number 423113162 (V.E.D.) and 433682494 – SFB 1459 (S.O.D), by the ANR MAESTRO project, Grant No. 18-CE24-0021 of the French Agence Nationale de la Recherche (A.A.), Labex NanoSaclay "SPICY" ANR–10-LABX-0035 (V.C.), and received financial support from the Horizon 2020 Framework Program of the European Commission under FET-Open grant agreement no. 899646 (k-NET) (A.A.). H. M. thanks L. Thevenard for useful discussions.

## Author contributions
H.M. performed the nanofabrication, measurements, and data analysis. B.D. performed measurements and data analysis. D.G. grew and characterized the films. A.E.K. performed data analysis. R.L., V.C., and P.B. contributed to the design and implementation of the research. A.A., S.O.D. and V.E.D. formulated the experimental approach and supervised the project. All authors co-wrote the manuscript.

## Funding

## Competing interests
The authors declare no competing interests.
