## [Peer Review File · Nature Communications]

True amplification of spin waves in magnonic nano-waveguidesREVIEWER COMMENTS

Reviewer #1 (Remarks to the Author):

In this manuscript, the author employs Bi: YIG, a material possessing perpendicular magnetic anisotropy, as a platform for magnon transport. By harnessing the SOT effect between the heavy metal Pt and ferrimagnetic materials, the author enhances the propagation of spin waves. Moreover, you capture the precise amplification signal within a specific time domain interval of spin wave propagation (transient regime and saturation regime) and delves into an in-depth discussion of the underlying physical mechanism behind this amplification.

The manuscript holds considerable significance in unveiling the enhanced transmission of spin waves through SOT. However, in order to enhance the overall coherence of the logical structure of the article, further explanations are required in the following aspects.

Main comments

1. In the device structure presented in this manuscript (Fig1.a), the excitation antenna is in direct proximity to the Pt layer, which raises concerns about potential galvanic interactions between the two. Specifically, the current flowing through Pt could directly influence the excitation of spin waves by being conducted to the antenna. To address this issue, it is important to understand how the author mitigated this phenomenon. Did you explore alternative measures such as applying a reverse direction of direct current?
2. Is the increasing process of current a continuous process, have you considered the process from small to large and then small?
3. What is the rationale behind selecting a 30° excitation angle for the external field? Have any experiment been conducted to investigate the correlation between angles?
4. In your explanation regarding the range of 139-154, the amplification of spin waves is attributed to the compensation of damping effects. Does SOT change the damping of Bi:YIG or generate new magnons? The related physical mechanism can be elucidated more clearly.
5. Did the author achieve spin wave excitation with low external field strength? Is there any excitation between 1-2 GHz?
6. Why is there no discussion about the impact of current exceeding 1.4mA on the spin wave? Is the amplification effect observed at 1.4mA considered the most optimal?

Additional comments

7. By what formula is the damping parameter of Bi:YIG/Pt obtained?
8. Why is the cell size setting of MuMax3 $10*10*10$, is it smaller than the spin diffusion length of the material? Have you evaluated the spin diffusion length of Bi:YIG?

Reviewer #2 (Remarks to the Author):

Dear Editor, Dear Authors,

In short: I believe that the paper is suitable for Scientific Reports. The manuscript is scientifically sound, the data is well explained and of high quality - but the manuscript lacks significance which would warrant publication in Nature Communications. Many pieces of the manuscript have already been reported elsewhere. The authors solve the problem of

nonlinear interaction and the presence of auto-oscillations for one particular geometry with a very specific magnetic field configuration. It's a nice finding but I feel it will only be of interest for a very small group of researchers.

Main criticism:

Regarding the Title: I find the emphasis on "True amplification" not appropriate. In the end, the type of information processing the authors are referring to in future applications are based on spin-wave interference where information would be encoded in the phase of spin waves. The authors did not show how the signal to noise ratio is affected by their type of SOT-amplification and the impact of the magnetic field applied under a very specific angle if it comes to informations (signals) encoded in the phase of spin waves. One could say the authors show an increase of the intensity of the carrier-wave, but from this one cannot conclude on the amplification of a signal unless further evidence is given

Regarding the current density: The authors write: "Note that 1.4 mA corresponds to a current density of only 4×10^{11} A/m². A significantly higher current could therefore be considered for future devices." This is misleading. First, the current density is already quite high and might cause substantial heating of the ferromagnetic material, which will hamper the use in applications.

More importantly, a current of 1.4mA in a Pt wire with such small dimensions (Resistance likely to be around 700Ohm) needs a Voltage of 1V. Going any higher in voltage will cause electrical breakdown of the insulating layer which is between the Pt stripe and the Au antenna for spin-wave excitation.

Note: The authors did not mention how the insulating layer (which is visible in Fig. 1b as the shaded square) was prepared. If the insulating layer is thinner than 100nm the electric field between the microwave line and the Pt stripe is already larger than 10MV/m and electrical breakdown is likely.

Regarding technological relevance: The authors mention in the conclusion that clocked operation is fully compatible with CMOS operation. However, the material used as spin-wave waveguide (BiYIG) requires annealing of the substrate at 700 degree Celsius and deposition at 420 degree Celsius. Hence, no CMOS compatibility at all.

Minor comments/suggestions:

Fig 1b: What the shaded, square shaped area with approx. $25 \times 25 \mu\text{m}^2$ size?

Fig. 4: For the sake of completeness the authors could add here the angle (30 degree?) in the figure or in the caption.

Fig 5b: Maybe use kappa in the y-axis label as introduced in Equation (1)

Reviewer #3 (Remarks to the Author):

The manuscript entitled "True amplification of spin waves in magnonic nano-waveguides" by

H. Merbouche et al. is devoted to the amplification of a spin wave signal – one of the main problems in the field of magnonics along with thermal stability, the need for a small-size magnetizing system, low-cost materials with low spin wave damping, etc. For spin wave amplification, authors use the effect of spin-orbit torque in layered structures. Before, this effect did not allow to achieve an increase in the spin wave amplitude with distance because of the appearance of magnetization auto-oscillations and corresponding nonlinear scattering of spin wave signal. Authors found a way to overcome arising limitations using combination of nonlinearity control and temporal separation that is the key noteworthy result of the presented manuscript. Therefore, I think the described findings will be of great interest to researchers in the field of magnonics, spintronics and related fields.

Used methodology certainly meets the expected standards in the field of magnonics, it is valid and robust. Pulsed laser deposition method for sample fabrication allows to get yttrium iron garnet films of good quality. The micro-focus Brillouin light scattering (BLS) spectroscopy is one of the most appropriate methods to directly observe spin wave propagation. MuMax3 is well-known and widely used software for micromagnetic simulations.

The manuscript references previous literature appropriately, described results do not contradict to known physics and spin wave behavior. The only thing that the reference [39] in the manuscript should tell us about typical current densities in magnonic devices according to the context: “Note that 1.4 mA corresponds to a current density of only 4×10^{11} A/m². A significantly higher current could therefore be considered for future devices³⁹.” The work [39] is devoted to relativistic kinematics of a magnetic soliton, and I did not find information about currents.

I have several comments that may improve the manuscript after a minor revision:

1. From the figure 1 (a) and the description of studied sample fabrication, it follows that there was no dielectric layer between the Pt layer and Au antenna. It would be nice to describe why. Probably, authors forgot to show this layer, otherwise part of a signal from antenna would go to the Pt layer, and part of a pulse applied to the Pt layer would go to the antenna.
2. Authors write that they simulated the spin-wave dispersion by micromagnetic simulations. For both used directions of applied field, it will be useful to show the dispersion itself in any figure for readers to understand what kind of spin wave was used in the experiment (surface, forward volume or backward volume spin wave) and which mode of spin waves as well (fundamental mode, width mode, or anisotropy mode). For the studied sample with huge anisotropy and an oblique direction of the applied field, it is not trivial to imagine the dispersion view.
3. In the description of spin wave excitation, it needs to point out the power of exciting signal, if it was below or higher the threshold values for three- and four-magnon decays. In another words, was the spin wave excited in linear or nonlinear regime?
4. Electrical current applied to the Pt layer rises its temperature and, thus, the temperature of the underlying Bi:YIG waveguide. For used dimensions of Pt layer and table values of Pt parameters (resistivity 9.81×10^{-8} Ohm \times m, specific heat capacity 134 J/(kg \times K), density 21.45×10^3 kg/m³), the used current value of 1.4 mA should increase the temperature as $dT/dt = 7.4$ K/ns according to expressions, for example, in [Phys. Rev. B, V. 84. 054437 (2011)]. This implies that a pulse with the duration of $dt=200$ ns will increase the temperature by $dT=1480$ K. The substrate and contacts will definitely absorb significant fraction of the heat and the temperature will not increase so high but it should be considerable anyway. Therefore, the spin wave spectrum can considerably change through the temperature dependence of magnetization and/or anisotropy. I think that the authors should discuss the heating influence in their manuscript.
5. Were BLS-curves in Figure 2 (b,c) measured when only a DC pulse was applied to Pt layer without excitation by Au antenna? While in Figure 2 (a) both pulses were applied?

6. Why the auto-oscillation frequency increases with I at $\theta = 0$? I expect that dynamic demagnetization should decrease the effective magnetization and, thus, shift the spin wave spectrum to lower frequencies. Is this also influence of perpendicular magnetic anisotropy?
7. For the simulations, two angles β of the magnetization precession cone were used, namely, $\beta = 10^\circ$ and 0.1° . How can we define the value of β for the given value of current I applied to Pt? The frequency shift is 20 MHz in the simulation at $\theta = 0$ (Figure 3, a), while the shift is around 200 MHz in the experiment (Figure 2, b) though $\beta = 10^\circ$ is a very high value implying dynamic part of magnetization $m = M \cdot \sin(10^\circ) \approx 260$ G. What is the reason for this discrepancy between experiment and simulation?
8. In the discussion about angular dependences of the ellipticity of the magnetization precession in the center and at the edge of the nano-waveguide, it is assumed that this is about zero wave number, at the frequencies of auto-oscillation peak. Does such relation for ellipticity retain for the frequencies outside the auto-oscillation peak? If so, why there is no auto-oscillation at these frequencies?
9. For Figure 4(a), I recommend changing the color of the curve for $I=1.0$ mA from blue to another, as it is easy to think that the blue squares and blue line in Figure 4(c) correspond to the blue line in Figure 4(a). At least, I had such a misconception when I first looked at this picture.
10. How was the group velocity measured for the Figure 5?
11. I do not know what letter "s" in the abbreviation "sGGG" stands for in the "Micro-focus BLS measurements" section. Is it any special gallium gadolinium garnet?
12. In the section "Methods. Micromagnetic simulations", I would like to see more details. How many initial spatially-periodic deflections of magnetization (how many points for the wave numbers and with which step) were used? What time sampling and duration were used for each deflection? What does the analysis of the free dynamics of magnetization mean? Is it fast Fourier transform?

Best regards,
Valentin Sakharov,
research fellow at Kotelnikov IRE RAS

Response to Reviewer #1

The Reviewer writes:

In this manuscript, the author employs Bi: YIG, a material possessing perpendicular magnetic anisotropy, as a platform for magnon transport. By harnessing the SOT effect between the heavy metal Pt and ferrimagnetic materials, the author enhances the propagation of spin waves. Moreover, you capture the precise amplification signal within a specific time domain interval of spin wave propagation (transient regime and saturation regime) and delves into an in-depth discussion of the underlying physical mechanism behind this amplification.

The manuscript holds considerable significance in unveiling the enhanced transmission of spin waves through SOT. However, in order to enhance the overall coherence of the logical structure of the article, further explanations are required in the following aspects.

Reply:

We thank the Reviewer for the positive evaluation of our work and the constructive comments aimed at the improvement of our manuscript. We hope that the Reviewer will find our answers and revisions satisfactory.

The Reviewer writes:

Main comments

1. In the device structure presented in this manuscript (Fig1.a), the excitation antenna is in direct proximity to the Pt layer, which raises concerns about potential galvanic interactions between the two. Specifically, the current flowing through Pt could directly influence the excitation of spin waves by being conducted to the antenna. To address this issue, it is important to understand how the author mitigated this phenomenon. Did you explore alternative measures such as applying a reverse direction of direct current?

Reply:

We are sorry for omitting the important details about the structure of the sample in the initial manuscript. In fact, the antenna is separated from the Pt layer by a 50 nm thick layer of SiO₂, so that there is no direct electrical contact between them.

Addressing the Reviewer's question, we have added information about the insulating layer on page 4 of the revised manuscript and in the Methods section.

The Reviewer writes:

2. Is the increasing process of current a continuous process, have you considered the process from small to large and then small?

Reply:

In fact, all measurements were performed in pulsed regime, i.e. the current was periodically applied during short time intervals (200-500 ns) and switched off for the rest of the period duration (≥ 4500 ns). This process cannot be considered as continuous increase of the current, since the current is switched on and off (large/small) many times during one measurement. The used pulsed regime significantly minimizes the Joule heating of the sample by the current, which otherwise could have indeed induced hysteretic effects.

To address the Reviewer's question, we have commented on this in the Methods section in the revised manuscript.

The Reviewer writes:

3. What is the rationale behind selecting a 30° excitation angle for the external field? Have any experiment been conducted to investigate the correlation between angles?

Reply:

The angle of 30° was selected to minimize nonlinear frequency shift and precession ellipticity, which is necessary to suppress detrimental nonlinear magnon interactions in a narrow waveguide. This angle was first found experimentally by systematically analyzing the angular dependence of the nonlinear frequency shift (Fig. 2b and 2c). Its value was further justified based on the results of micromagnetic simulations (Fig. 3), which provide insight into what makes this angle optimal for ellipticity minimization.

To address the Reviewer's question, we have mentioned the reason for the selection of 30° at the beginning of the experimental section (description of Fig. 1 in the main text) (page 5 of the revised manuscript) and referred to the detailed explanations given in the following sections.

The Reviewer writes:

4. In your explanation regarding the range of 139-154, the amplification of spin waves is attributed to the compensation of damping effects. Does SOT change the damping of Bi:YIG or generate new magnons? The related physical mechanism can be elucidated more clearly.

Reply:

We agree with the Reviewer that this should be explained in more detail. The processes discussed in the paragraph mentioned by the Reviewer are associated with the effect of the damping compensation by SOT. To achieve an exponential increase in the intensity of spin waves during propagation, this effect must be strong enough to "reverse" the damping, i.e., the effective damping must become negative. Under these conditions, the system becomes unstable. This inevitably leads to the development of magnetic auto-oscillations at a frequency corresponding to the dynamic state with the lowest damping. Although these auto-oscillations have a frequency that is different from the frequency of spin waves to be amplified, their large amplitude leads to nonlinear scattering of spin waves excited by the antenna and prevents their amplification by SOT. In our work, we show how this detrimental nonlinear scattering can be suppressed. Additionally, we use the signal from auto-oscillations to characterize the nonlinear frequency shift at different angles of the static magnetic field.

Addressing the Reviewer's question, we have explained this in more detail on page 7 of the revised manuscript.

The Reviewer writes:

5. Did the author achieve spin wave excitation with low external field strength? Is there any excitation between 1-2 GHz?

Reply:

In this work, we did not consider the particular case of low frequencies at low field strengths. However, our previous experiments with BiYIG/Pt (see, e.g., Ref. 38) showed that the anti-damping effects of SOT generally become stronger with decreasing frequency/field.

The Reviewer writes:

6. Why is there no discussion about the impact of current exceeding 1.4mA on the spin wave? Is the amplification effect observed at 1.4mA considered the most optimal?

Reply:

The reason for the limitation of the current at 1.4 mA is purely technical. From preliminary measurements with similar devices, we found that during long-term measurements, the devices can become damaged at currents 1.5-1.8 mA, likely due to electromigration effects at high current densities. Therefore, in our main experiments, we did not increase the current above 1.4 mA to ensure that the device under study does not degrade in the course of many-weeks measurements, including regimes where current is applied continuously and/or in relatively long pulses. We believe that the current density can be increased beyond the maximum value used in our experiments. However this requires further optimization of experimental samples.

Addressing the Reviewer's question, we have added a comment about the maximum current on page 6 of the revised manuscript.

The Reviewer writes:

Additional comments

7. By what formula is the damping parameter of Bi:YIG/Pt obtained?

Reply:

Addressing the Reviewer's comment, we have added the formula on page 6 of the revised manuscript.

The Reviewer writes:

8. Why is the cell size setting of MuMax3 $10*10*10$, is it smaller than the spin diffusion length of the material? Have you evaluated the spin diffusion length of Bi:YIG?

Reply:

We use micromagnetic simulations only to obtain the dispersion and nonlinear characteristics of spin waves in the BiYIG waveguide. We do not model the injection of spin current or spin-transfer torque effects. Therefore, the spin diffusion length does not enter our simulations as a parameter. The cell size of 10 nm is chosen to be smaller than the exchange length of YIG, which is approximately equal to 18 nm.

In response to the Reviewer's question, we have commented on the choice of the cell size in the Methods section.

Response to Reviewer #2

The Reviewer writes:

Dear Editor, Dear Authors,

In short: I believe that the paper is suitable for Scientific Reports. The manuscript is scientifically sound, the data is well explained and of high quality - but the manuscript lacks significance which would warrant publication in Nature Communications. Many pieces of the manuscript have already been reported elsewhere.

Reply:

We thank the Reviewer for recognizing the "high quality" of our research. We respectfully disagree with the Reviewer that our results lack significance. We hope that our response to the Reviewer's comments will help to change the Reviewer's opinion.

The Reviewer writes:

The authors solve the problem of nonlinear interaction and the presence of auto-oscillations for one particular geometry with a very specific magnetic field configuration.

Reply:

We would like to emphasize that we not only solve the problem for "one particular geometry with a very specific magnetic field configuration", we demonstrate an unprecedented methodology and reveal generalized conditions for any magnonic system based on new insight of the time-dependent non-linear physics of magnetic materials. In fact, our approach can be extended to a large class of magnetic materials and device geometries.

The Reviewer writes:

It's a nice finding but I feel it will only be of interest for a very small group of researchers.

Reply:

The problem of efficient amplification of spin waves at the nanoscale is one of the most important long-standing problems in the field of magnonics. Therefore, its solution should be of interest for all researchers working in the field.

The Reviewer writes:

Main criticism:

Regarding the Title: I find the emphasis on "True amplification" not appropriate. In the end, the type of information processing the authors are referring to in future applications are based on spin-wave interference where information would be encoded in the phase of spin waves. The authors did not show how the signal to noise ratio is affected by their type of SOT-amplification and the impact of the magnetic field applied under a very specific angle if it comes to informations (signals) encoded in the phase of spin waves. One could say the authors show an increase of the intensity of the carrier-wave, but from this one cannot conclude on the amplification of a signal unless further evidence is given

Reply:

Although we agree with the Reviewer that "future applications are based on spin-wave interference where information would be encoded in the phase of spin waves", we would like to emphasize that in a high-discrimination interference experiment, the interfering waves must have similar input amplitudes. Additionally, the amplitudes of the waves must be much higher than the noise of the detectors used. Therefore, the ability to control the amplitude of spin waves is not less important than the ability to control their phase. In our work, we propose a general solution for the long-standing problem of amplitude amplification. We agree that the use of this solution in commercial end devices will require optimization of the signal-to-noise ratio. However, optimizing such parameters is an engineering task that clearly falls beyond the scope of our physical study.

The Reviewer writes:

Regarding the current density: The authors write: "Note that 1.4 mA corresponds to a current density of only 4×10^{11} A/m². A significantly higher current could therefore be considered for future devices." This is misleading. First, the current density is already quite high and might cause substantial heating of the ferromagnetic material, which will hamper the use in applications.

Reply:

We would like to emphasize that the current density of 4×10^{11} A/m² is of the same order of magnitude (or even smaller) as the current density required to operate most spintronics devices, such as MRAMs, which have almost unlimited endurance. We agree that the heating is a common problem in spintronics. However, it does not prevent the development of this research field. In our experiments, we limit the current density to a relatively small value to ensure the stability of our experimental samples in long-term measurements, including regimes where current is applied continuously and/or in relatively long pulses. Further optimization of samples and operating regimes can definitely help to significantly increase the operating current density.

The Reviewer writes:

More importantly, a current of 1.4mA in a Pt wire with such small dimensions (Resistance likely to be around 700Ohm) needs a Voltage of 1V. Going any higher in voltage will cause electrical breakdown of the insulating layer which is between the Pt stripe and the Au antenna for spin-wave excitation.

Reply:

We would like to emphasize that our approach does not require applying voltage across the insulating layer. If electrical breakdown between the Pt electrode and the microwave antenna becomes a limiting factor (unlikely), it can be easily eliminated by using DC blocking capacitors in the microwave path or by increasing the insulating layer thickness. Here, unlike conventional MOSFETs, the dielectric layer does not play any role in the rf electrical characteristics of the devices.

The Reviewer writes:

Note: The authors did not mention how the insulating layer (which is visible in Fig. 1b as the shaded square) was prepared. If the insulating layer is thinner than 100nm the electric field between the microwave line and the Pt stripe is already larger than 10MV/m and electrical breakdown is likely.

Reply:

We thank the Reviewer for pointing out this omission. In the revised manuscript, we provide details about the insulating layer on page 4 and in the Methods section.

We note that SiO₂ used as the insulating layer material has a tabulated breakdown field of 10 MV/cm. Even if we assume that this value is ten times smaller for our 50-nm thick film (1 MV/cm), it is still much larger than the electric field in our devices (0.2 MV/cm). Therefore, even without the use DC blocking capacitors, the risk of electrical breakdown in our devices is negligible.

The Reviewer writes:

Regarding technological relevance: The authors mention in the conclusion that clocked operation is fully compatible with CMOS operation. However, the material used as spin-wave waveguide (BiYIG) requires annealing of the substrate at 700 degree Celsius and deposition at 420 degree Celsius. Hence, no CMOS compatibility at all.

Reply:

The statement about CMOS compatibility mentioned by the reviewer only refers to the electronics necessary to perform the amplification. A square-shaped pulse of 100 ns with an amplitude of 1V does not require any special additions to conventional CMOS circuitry because there is no need for voltages significantly higher than the standard CMOS voltages. Regarding materials, the Reviewer is correct that material compatibility is still an open problem for YIG-

based magnonics. However, this does not exclude CMOS compatibility. The most advanced technological research is exploring the path of wafer bonding of magnonic circuits and CMOS control electronics. More importantly, our amplification approach is not specific to YIG. It can also be used with CMOS-compatible magnetic materials with perpendicular magnetic anisotropy.

Minor comments/suggestions:

Fig 1b: What the shaded, square shaped area with approx. $25 \times 25 \mu\text{m}^2$ size?

Reply:

This area is the insulating layer. As mentioned above, we have added its description in the revised manuscript.

Fig. 4: For the sake of completeness the authors could add here the angle (30 degree?) in the figure or in the caption.

Reply:

The caption of Fig. 4 already indicates that the data are obtained at $\theta = 30^\circ$.

Fig 5b: Maybe use kappa in the y-axis label as introduced in Equation (1)

Reply:

Following the Reviewer's recommendation, in the revised manuscript, we use the symbol "kappa" in addition to the text description in the y-axis labels in Figs. 2a and 5b.

Response to Reviewer #3

The Reviewer writes:

The manuscript entitled "True amplification of spin waves in magnonic nano-waveguides" by H. Merbouche et al. is devoted to the amplification of a spin wave signal – one of the main problems in the field of magnonics along with thermal stability, the need for a small-size magnetizing system, low-cost materials with low spin wave damping, etc. For spin wave amplification, authors use the effect of spin-orbit torque in layered structures. Before, this effect did not allow to achieve an increase in the spin wave amplitude with distance because of the appearance of magnetization auto-oscillations and corresponding nonlinear scattering of spin wave signal. Authors found a way to overcome arising limitations using combination of nonlinearity control and temporal separation that is the key noteworthy result of the presented manuscript. Therefore, I think the described findings will be of great interest to researchers in the field of magnonics, spintronics and related fields.

Used methodology certainly meets the expected standards in the field of magnonics, it is valid and robust. Pulsed laser deposition method for sample fabrication allows to get yttrium iron garnet films of good quality. The micro-focus Brillouin light scattering (BLS) spectroscopy is one of the most appropriate methods to directly observe spin wave propagation. MuMax3 is well-known and widely used software for micromagnetic simulations.

Reply:

We thank the Reviewer for the positive evaluation of our work and the constructive comments.

The Reviewer writes:

The manuscript references previous literature appropriately, described results do not contradict to known physics and spin wave behavior. The only thing that the reference [39] in the manuscript should tell us about typical current densities in magnonic devices according to the context: “Note that 1.4 mA corresponds to a current density of only 4×10^{11} A/m². A significantly higher current could therefore be considered for future devices³⁹.” The work [39] is devoted to relativistic kinematics of a magnetic soliton, and I did not find information about currents.

Reply:

We agree with the Reviewer, that this needs to be clarified. In fact, the work Ref. 39 reports reliable operation of BiYIG/Pt structures at current densities of more than 10^{12} A/m², which is much larger than the current densities used in our work. We have rewritten the corresponding paragraph (page 6 in the revised manuscript) to make this clearer.

The Reviewer writes:

I have several comments that may improve the manuscript after a minor revision:

1. From the figure 1 (a) and the description of studied sample fabrication, it follows that there was no dielectric layer between the Pt layer and Au antenna. It would be nice to describe why. Probably, authors forgot to show this layer, otherwise part of a signal from antenna would go to the Pt layer, and part of a pulse applied to the Pt layer would go to the antenna.

Reply:

We thank the Reviewer for bringing this omission to our attention. In the revised manuscript, we have indicated that the Au antenna is isolated from the Pt layer by a 50 nm thick layer of SiO₂ (page 4 and the Methods section).

The Reviewer writes:

2. Authors write that they simulated the spin-wave dispersion by micromagnetic simulations. For both used directions of applied field, it will be useful to show the dispersion itself in any figure for readers to understand what kind of spin wave was used in the experiment (surface, forward volume or backward volume spin wave) and which mode of spin waves as well (fundamental mode, width mode, or anisotropy mode). For the studied sample with huge anisotropy and an oblique direction of the applied field, it is not trivial to imagine the dispersion view.

Reply:

We agree with the Reviewer that information about spin-wave dispersion is important. Complying with the Reviewer's request we have shown the calculated dispersion curves in new Fig. 3a and discuss them on pages 7-8 of the revised manuscript.

The Reviewer writes:

3. In the description of spin wave excitation, it needs to point out the power of exciting signal, if it was below or higher the threshold values for three- and four-magnon decays. In another words, was the spin wave excited in linear or nonlinear regime?

Reply:

Following the Reviewer's suggestion, we have specified the excitation power (0.1 mW) in the description of the experiment (page 4 of the revised manuscript). We have also indicated that this power corresponds to the linear excitation regime. In our experiments, we always prove the latter by measuring the intensity of dynamic magnetization near the antenna as a function of the excitation power and choose the experimental power so that it is at least an order of magnitude lower than the threshold, at which the response becomes nonlinear.

The Reviewer writes:

4. Electrical current applied to the Pt layer rises its temperature and, thus, the temperature of the underlying Bi:YIG waveguide. For used dimensions of Pt layer and table values of Pt parameters (resistivity 9.81×10^{-8} Ohm \times m, specific heat capacity 134 J/(kg \times K), density 21.45×10^3 kg/m³), the used current value of 1.4 mA should increase the temperature as $dT/dt = 7.4$ K/ns according to expressions, for example, in [Phys. Rev. B, V. 84. 054437 (2011)]. This implies that a pulse with the duration of $dt=200$ ns will increase the temperature by $dT=1480$ K. The substrate and contacts will definitely absorb significant fraction of the heat and the temperature will not increase so high but it should be considerable anyway. Therefore, the spin wave spectrum can considerably change through the temperature dependence of magnetization and/or anisotropy. I think that the authors should discuss the heating influence in their manuscript.

Reply:

Following the Reviewer's suggestion, we have discussed heating effects in the Methods section. We estimated the heating of the sample based on measurements of its electrical resistance on current strength (new Supplementary Fig. 1). According to these data obtained in the *continuous-current* regime, the maximum increase of the temperature at $I = 1.4$ mA is 130 K. As the Reviewer noted, this value is significantly smaller than that obtained from simple estimates due to efficient heat dissipation through the single crystalline substrate, SiO₂ capping layer, and gold electrodes. We note that this value is in good agreement with the values obtained for similar structures by other researchers (see, e.g., new Ref. 44). Additionally, in the *pulsed* mode used in our main measurements, the temperature increase is expected to be even smaller.

We would like to note separately that the effects of the heating on the spin-wave spectrum are similar to the nonlinear effects associated with an increase in the amplitude of precession. Both of them result in the reduction of the projection of the magnetization vector onto the precession axis and lead to a frequency shift. Therefore, under conditions of zero nonlinear shift, the heating does not significantly modify the dispersion spectrum.

The Reviewer writes:

5. Were BLS-curves in Figure 2 (b,c) measured when only a DC pulse was applied to Pt layer without excitation by Au antenna? While in Figure 2 (a) both pulses were applied?

Reply:

This is correct. To make this clearer, we have indicated in the caption of Fig. 2, that the data shown in panels b and c are obtained without applying microwave pulses to the antenna.

The Reviewer writes:

6. Why the auto-oscillation frequency increases with I at $\theta = 0$? I expect that dynamic demagnetization should decrease the effective magnetization and, thus, shift the spin wave spectrum to lower frequencies. Is this also influence of perpendicular magnetic anisotropy?

Reply:

This is correct. The reason for the positive nonlinear frequency shift is perpendicular magnetic anisotropy. A decrease in the projection of the magnetization onto the precession axis leads to a positive shift if the effective anisotropy field is larger than the saturation magnetization of the film, which is the case for our samples.

The Reviewer writes:

7. For the simulations, two angles β of the magnetization precession cone were used, namely, $\beta = 10^\circ$ and 0.1° . How can we define the value of β for the given value of current I applied to Pt? The frequency shift is 20 MHz in the simulation at $\theta = 0$ (Figure 3, a), while the shift is around 200 MHz in the experiment (Figure 2, b) though $\beta = 10^\circ$ is a very high value implying dynamic part of magnetization $m = M \cdot \sin(10^\circ) \approx 260$ G. What is the reason for this discrepancy between experiment and simulation?

Reply:

We agree with the Reviewer that this needs to be clarified. As the Reviewer correctly noted above, the frequency shift is also determined by an increase in the temperature of the sample due to heating by electrical current. This additional contribution is the reason why the absolute value of the nonlinear shift observed experimentally at $\theta = 0$ is significantly larger than the values obtained from simulations. Since the two contributions are difficult to distinguish, we do not compare the absolute values of the shift, but only analyze its angular dependence. In response to the Reviewer's question, we have added clarifications on page 8 of the revised manuscript.

The Reviewer writes:

8. In the discussion about angular dependences of the ellipticity of the magnetization precession in the center and at the edge of the nano-waveguide, it is assumed that this is about zero wave number, at the frequencies of auto-oscillation peak. Does such relation for ellipticity retain for the frequencies outside the auto-oscillation peak? If so, why there is no auto-oscillation at these frequencies?

Reply:

The relation for the ellipticity is valid over a wide range of wave vectors, not just for the zero wave vector. The fundamental reason for single-mode auto-oscillations is the Bose nature of magnons. Fundamentally, an ensemble of non-interacting Bose-particles (magnons) tends to form a single overpopulated state (Bose-Einstein condensation). In most magnetic systems, this process is destroyed by strong magnon-magnon interactions caused by the dipolar coupling (see, e.g., Nature Commun. 12, 6541 (2021)). In our work, we use perpendicular magnetic anisotropy to compensate the dipolar effects, which leads to the suppression of magnon-magnon interactions. As a result, our system exhibits single-mode auto-oscillations at the frequency corresponding to the lowest-energy magnon state.

The Reviewer writes:

9. For Figure 4(a), I recommend changing the color of the curve for $I=1.0$ mA from blue to another, as it is easy to think that the blue squares and blue line in Figure 4(c) correspond to the blue line in Figure 4(a). At least, I had such a misconception when I first looked at this picture.

Reply:

As requested by the Reviewer, we have changed the color of the $I=1$ mA curve from blue to black.

The Reviewer writes:

10. How was the group velocity measured for the Figure 5?

Reply:

The group velocity was determined from the spatio-temporal shift of the edge of the spin-wave pulse similar to Fig. 1c. We have indicated this on page 10 of the revised manuscript. Since the temporal resolution of BLS is better than 1 ns, we can detect the arrival of a spin-wave packet at a given distance from the antenna with very good accuracy. We observe a linear dependence of the arrival time on the distance traveled. The slope of this dependence provides information about the group velocity.

The Reviewer writes:

11. I do not know what letter “s” in the abbreviation “sGGG” stands for in the “Micro-focus BLS measurements” section. Is it any special gallium gadolinium garnet?

Reply:

Addressing the Reviewer’s question, we have clarified in the Methods section of the revised manuscript that sGGG is the substituted Gallium Gadolinium Garnet. Substrate suppliers dope the pristine GGG ($\text{Gd}_3\text{Ga}_5\text{O}_{12}$) using Ca, Mg, Zr atoms to increase the unit cell parameter from 1.2376 nm to larger values without changing the crystal symmetry. The important parameter is the unit cell parameter we used, which is 1.2497 nm. This information was indeed missing. Now it is given in the Methods section.

The Reviewer writes:

12. In the section “Methods. Micromagnetic simulations”, I would like to see more details. How many initial spatially-periodic deflections of magnetization (how many points for the wave numbers and with which step) were used? What time sampling and duration were used for each deflection? What does the analysis of the free dynamics of magnetization mean? Is it fast Fourier transform?

Reply:

Following the Reviewer’s suggestion, we have included all requested information in the Methods section. We have also corrected the length of the waveguide used in the simulations. It was mistakenly given as 20 μm , but was 10 μm .

REVIEWER COMMENTS

Reviewer #1 (Remarks to the Author):

The manuscript is improved and most of my comments adequately addressed. I would like to support its publication on Nature Communications. However, the physical mechanism still needs to be further explained and analyzed.

Some comments

1.(p. 7, lines 153-157) It is written: "Although these auto-oscillations have a frequency different from the frequency of spin waves to be amplified, the nonlinear scattering of these waves from intense auto-oscillations prevents their amplification". The nonlinear scattering appears to depend on the ellipticity of the magnetization precession. Please consider elaborate further on the nonlinear scattering and its relation to ellipticity.

2.(p. 7, lines 165) The author mentioned the auto-oscillation frequency changes is due to the interplay between the effects of PMA and the dipolar demagnetizing fields. It seems more likely that PMA and static magnetic field decided the direction of the stable magnetic moment, and spin current and damping dissipation make the magnetic moment precesses in an elliptical trajectory about the direction of the effective field. Please consider drawing a diagram of the torque in different states during the precession of the magnetic moment.

3.The discussion on the reason of nonlinear frequency shift with the change of θ would be also important.

Reviewer #2 (Remarks to the Author):

I do not feel that the authors' reply is addressing my comments in an satisfactory manner. The paper is interesting for the magnonics community. The authors write this in their rebuttal themselves. But the magnonics community is a very specialized subsection of magnetism and an even smaller subsection of condensed matter physics. Hence, not important for a larger group.

My impression, that the paper is solid, but specialized for a particular geometry, ist still unchanged. The authors claim that their approach can also be used for CMOS compatible materials is misleading., CMOS compatible materials would require orders of magnitude more magnetic field to achieve the studied geometry with minimized nonlinear effects.

Regarding the signal amplification itself: A detailed discussion of the signal to noise ratio before and after the amplification stage would be a true measure for spin-wave amplification. Otherwise it is not clear of the increased signal intensity seen in BLS is simply an incoherent background. The authors have one of the most advanced BLS setups in the world including phase resolution. Phase-resolved measurements would be true evidence for spin wave amplification

Furthermore, the reply regarding the electrical breakdown of the device is incorrect. As I calculated in my first review, the a current of 1.4mA in the Pt layer (700Ohm resistance) requires a Voltage of 1V. With a thickness of the insulation layer of 50nm this results in an electric field of 20MV between the Pt layer and the antenna for the spin wave excitation.

Hence, field is reaching critical values. The authors mention other values for the electric field in their device, but they did not provide an explanation, how this values were derived.

Reviewer #3 (Remarks to the Author):

The authors responded to all my remarks on their manuscript. In my opinion, the revised manuscript can be published as is.

Response to Reviewer #1

The Reviewer writes:

The manuscript is improved and most of my comments adequately addressed. I would like to support its publication on Nature Communications. However, the physical mechanism still needs to be further explained and analyzed.

Reply:

We thank the Reviewer for supporting publication of our work in Nature Communications. As described in detail below, in the revised manuscript, we have thoroughly addresses all additional Reviewer's inquiries.

The Reviewer writes:

Some comments

1.(p. 7, lines 153-157) It is written: "Although these auto-oscillations have a frequency different from the frequency of spin waves to be amplified, the nonlinear scattering of these waves from intense auto-oscillations prevents their amplification". The nonlinear scattering appears to depend on the ellipticity of the magnetization precession. Please consider elaborate further on the nonlinear scattering and its relation to ellipticity.

Reply:

Complying with the Reviewer's request, we have added detailed explanation on the nonlinear scattering and its relation to ellipticity on pages 7-8 of the revised manuscript.

Nonlinear scattering is the exchange of energy between different spin waves caused by their nonlinear interaction. In the linear approximation, all magnon (spin-wave) eigenstates are orthogonal and independent of each other. However, with an increase in the amplitude of magnetization precession, nonlinear coupling mechanisms become active. The most efficient coupling mechanism is associated with the parametric interaction of spin waves, which requires the precession to be elliptical. As illustrated in the new Fig. 3a, the large-amplitude elliptical precession of the magnetization vector leads to the appearance of a component of the dynamic magnetization $m_{||}$, parallel to the precession axis. This component represents a periodic modulation of the parameter of the system – static magnetization. If the modulation amplitude is large enough, the modulation can lead to parametric excitation of pairs of spin waves whose frequencies and wave vectors are different from those of the initial spin wave. In other words, different spin waves become coupled.

In magnetic films, the ellipticity of the magnetization precession is due to the dynamic dipolar demagnetizing field h_d , which is antiparallel to the out-of-plane component of the dynamic magnetization (new Fig. 3a). In films with PMA (new Fig. 3b), the dipolar field h_d can be compensated by the effective field of the anisotropy h_a , which is oriented parallel to the out-of-plane component of the magnetization. This results in a decrease in the ellipticity. In the case of a narrow waveguide, the trajectory of the magnetization vector is additionally affected by in-plane demagnetizing fields caused by the lateral edges (see the results of simulations in Fig. 4c). By varying the angle of the static magnetic field θ , one can control the relative contributions of all these fields and minimize ellipticity and, as a consequence, nonlinear scattering.

The Reviewer writes:

2.(p. 7, lines 165) The author mentioned the auto-oscillation frequency changes is due to the interplay between the effects of PMA and the dipolar demagnetizing fields. It seems

more likely that PMA and static magnetic field decided the direction of the stable magnetic moment, and spin current and damping dissipation make the magnetic moment precesses in an elliptical trajectory about the direction of the effective field. Please consider drawing a diagram of the torque in different states during the precession of the magnetic moment.

Reply:

We completely agree with the Reviewer, that PMA and the static magnetic field decide the direction of the equilibrium orientation of the static magnetization. However, as described above, the balance between the *dynamic* anisotropy field and the *dynamic* demagnetizing fields also determines the ellipticity of the magnetization precession. As suggested by the Reviewer, we have shown diagrams illustrating these effects in new Fig. 3. We have also explained the effects of the anisotropy on the nonlinear frequency shift (see our reply to the comment 3 below). Separately, we note that the *damping-like* spin-orbit torque does not change the trajectory of magnetization precession. It only brings the magnetization in motion along a trajectory, which is determined by the dynamic fields in the system. The trajectory can be influenced by the *field-like* spin-orbit torque. However, this torque is known to be pronounced only in ultra-thin magnetic films with sub-nm thickness.

The Reviewer writes:

3.The discussion on the reason of nonlinear frequency shift with the change of θ would be also important.

Reply:

Complying with the Reviewer's request, we have added detailed explanations of the nonlinear frequency shift as Supplementary Note 1 and referenced them on page 7 of the revised manuscript.

The nonlinear frequency shift is a change in the frequency of magnetization precession due to a decrease in the static component of magnetization M_{ST} with increasing precession angle. The simplest way to understand the effects of anisotropy and angle of the static magnetic field on nonlinear frequency shift is to consider the frequency of the ferromagnetic resonance (FMR) for two limiting cases: in-plane (IP) magnetized film ($\theta = 0$) and out-of-plane (OP) magnetized film ($\theta = 90^\circ$). In the first case, the FMR frequency can be expressed as $\omega_{IP} = \gamma\sqrt{H_0(H_0 + 4\pi M_{eff})}$. In the second case, it is $\omega_{OP} = \gamma(H_0 - 4\pi M_{eff})$. Here γ is the gyromagnetic ratio and M_{eff} is the effective magnetization. In films with perpendicular magnetic anisotropy (see new Supplementary Ref. 1), $4\pi M_{eff} = 4\pi M_{ST} - H_a = (4\pi - N^a)M_{ST}$, where M_{ST} is the static component of the magnetization, $H_a = N^a M_{ST}$ is the effective field of the anisotropy, and N^a is the anisotropy constant.

In the case, if there is no anisotropy or the effective field of the anisotropy H_a is smaller than $4\pi M_{ST}$, $4\pi M_{eff}$ is positive. Under these conditions, a decrease in M_{ST} leads to a decrease in ω_{IP} (negative nonlinear frequency shift) and an increase in ω_{OP} (positive nonlinear frequency shift).

In the case $H_a > 4\pi M_{ST}$, which is the case in our experiments, $4\pi M_{eff}$ is negative. Under these conditions, a decrease in M_{ST} leads to an increase in ω_{IP} (positive nonlinear frequency shift) and a decrease of ω_{OP} (negative nonlinear frequency shift). Correspondingly, by changing the angle of the static magnetic field θ from 0 to 90 degrees, one can change the sign of the nonlinear frequency shift. Moreover, for a certain angle θ , one can achieve a situation where the nonlinear frequency shift vanishes.

Response to Reviewer #2

The Reviewer writes:

I do not feel that the authors' reply is addressing my comments in an satisfactory manner. The paper is interesting for the magnonics community. The authors write this in their rebuttal themselves. But the magnonics community is a very specialized subsection of magnetism and an even smaller subsection of condensed matter physics. Hence, not important for a larger group.

Reply:

We would like to draw the attention of the Reviewer that papers published by Nature Communications “aim to represent important advances of significance to *specialists within each field.*” Therefore, our results, which are of large interest to specialists in the field of magnonics and significantly advance this field, fully satisfy the criteria of the journal.

The Reviewer writes:

My impression, that the paper is solid, but specialized for a particular geometry, ist still unchanged. The authors claim that their approach can also be used for CMOS compatible materials is misleading., CMOS compatible materials would require orders of magnitude more magnetic field to achieve the studied geometry with minimized nonlinear effects.

Reply:

The statement by the Reviewer about “orders of magnitude more magnetic field” is not correct. Large fields are required to overcome the shape anisotropy in films caused by the dipole interaction. However, in films with perpendicular magnetic anisotropy tuned to compensate for dipolar demagnetizing field, orienting the magnetization at an angle to the surface does not require large fields. This applies to material used in our work and is equally applicable to CMOS-compatible materials. Such compensation was demonstrated, for example, for CoNi in Ref. 35.

The Reviewer writes:

Regarding the signal amplification itself: A detailed discussion of the signal to noise ratio before and after the amplification stage would be a true measure for spin-wave amplification. Otherwise it is not clear of the increased signal intensity seen in BLS is simply an incoherent background. The authors have one of the most advanced BLS setups in the world including phase resolution. Phase-resolved measurements would be true evidence for spin wave amplification

Reply:

We thank the Reviewer for clarifying the concern. As requested, we have performed phase-resolved measurements before and after the amplification stage (see new Fig. 7 and new chapter “Coherence of the amplification process”). The new data clearly show that the observed increase in the intensity is associated with the amplification of the coherent signal and is not due to an increase in the incoherent background. In other words, they “provide true evidence for spin wave amplification”.

We emphasize that the signal-to-noise ratio in these measurements is determined by the noise of the photon counter and, generally, increases with increasing measurement time. Therefore,

it cannot be used to quantify the actual signal-to-noise ratio in the system under study. However, from the data of Fig. 7, one clearly sees that the signal-to-noise ratio is excellent in our experiments.

The Reviewer writes:

Furthermore, the reply regarding the electrical breakdown of the device is incorrect. As I calculated in my first review, the a current of 1.4mA in the Pt layer (700Ohm resistance) requires a Voltage of 1V. With a thickness of the insulation layer of 50nm this results in an electric field of 20MV between the Pt layer and the antenna for the spin wave excitation. Hence, field is reaching critical values. The authors mention other values for the electric field in their device, but they did not provide an explanation, how this values were derived.

Reply:

With all respect, we do not understand which of the following statements requires further explanations:

- a) Electric field in our devices (as estimated by the Reviewer) $1 \text{ V} / 50 \times 10^{-9} \text{ m} = 20 \text{ MV/m}$.
- b) This electric field is *smaller* than the breakdown field in SiO₂ films (insulation material in our samples), which is more than 100 MV/m.
- c) If larger voltages are needed in future devices, the breakdown can be easily eliminated by using DC blocking capacitors in the microwave path or by increasing the insulating layer thickness.

Response to Reviewer #3

The Reviewer writes:

The authors responded to all my remarks on their manuscript. In my opinion, the revised manuscript can be published as is.

Reply:

We thank the Reviewer for the positive evaluation of our work and the recommendation to publish the revised manuscript in Nature Communications as is.

REVIEWERS' COMMENTS

Reviewer #1 (Remarks to the Author):

The authors have revised the manuscript according to my concerns. I would like to support its publication on Nature Communications in the current form.

Reviewer #2 (Remarks to the Author):

Dear Editor, Dear Authors,

Concerning the coherence: I very much appreciate the new measurements shown in Figure 7. I believe this was a missing piece to complete the story and improves the manuscript a lot.

Still, I believe that the manuscript only shows an incremental technological step of already existing knowledge, which the authors published recently in Nature Communications (see Ref. 35), and thus would much better fit into an applied physics journal. The advancement here is that the authors showed amplification for propagating spin waves and not just auto-oscillations in confined systems, but this required a very particular experimental geometry in a very specialized material which is not advancing the development of new technologies (see my next argument regarding CMOS compatibility). The authors refer to the journal's scope and mention the "importance and significance to specialists within each field". This is the reason why I considered the manuscript not fit for publication in Nature Communications, because I question the significance which would justify publication in such a high impact journal.

The reply by the authors that their approach is CMOS compatible is still misleading from my point of view. I mentioned in my second review, that large fields would be required in metallic systems to overcome the problem of nonlinear effects. The authors contradict my statement by citing their result in Ref. 35 where they showed the same mechanism for metallic materials with perpendicular magnetic anisotropy in devices made from CoNi. However, in their publication in Ref. 35 the authors study confined spin waves in a 500nm disc, which are not propagating and where no information transport could be realized. And still the applied magnetic field was 200mT.

Hence, my argument that the manuscript and data are nice and solid, but for a very specialized geometry, still holds and still doesn't allow for CMOS compatibility. If the authors showed the true amplification of propagating spin waves in samples similar as analyzed in Ref.35, then the claim of CMOS compatibility is justified, but still it would be an incremental achievement considering the results in Ref.35 and the present manuscript, because no new physical phenomena has been observed or a known mechanism has been utilized in a novel way.

Concerning the breakdown field: I calculated the electric field in SI-units (V/m), whereas the authors replied in V/cm, which I did not see in their first rebuttal letter. Hence, the authors are correct, the actual electric field is smaller than the breakdown field they mentioned. However, I believe they are wrong when stating that a breakdown could be avoided by inserting a dc block before the antenna for spin wave excitation. The dc-block protects the signal generator by integrating a capacitor in the signal line. But the ground line is still

connected to the ground potential, thus an electric field will build up between antenna and the magnetic material connected to the dc input.

Response to Reviewer #1

The Reviewer writes:

The authors have revised the manuscript according to my concerns. I would like to support its publication on Nature Communications in the current form.

Reply:

We thank the Reviewer for supporting publication of our work in Nature Communications in the current form.

Response to Reviewer #2

The Reviewer writes:

Concerning the coherence: I very much appreciate the new measurements shown in Figure 7. I believe this was a missing piece to complete the story and improves the manuscript a lot.

Reply:

We are pleased that the Reviewer finds the result of the additional measurements compelling.

The Reviewer writes:

Still, I believe that the manuscript only shows an incremental technological step of already existing knowledge, which the authors published recently in Nature Communications (see Ref. 35), and thus would much better fit into an applied physics journal. The advancement here is that the authors showed amplification for propagating spin waves and not just auto-oscillations in confined systems, but this required a very particular experimental geometry in a very specialized material which is not advancing the development of new technologies (see my next argument regarding CMOS compatibility). The authors refer to the journal's scope and mention the "importance and significance to specialists within each field". This is the reason why I considered the manuscript not fit for publication in Nature Communications, because I question the significance which would justify publication in such a high impact journal.

Reply:

We are very sorry that the Reviewer does not recognize that the problem of efficient amplification of spin waves at the nanoscale is one of the most important long-standing problems in the field of magnonics. With all respect, we firmly believe that this is the case and that the solution of this problem reported in our work represents a significant advance in the field.

The Reviewer writes:

The reply by the authors that their approach is CMOS compatible is still misleading from my point of view. I mentioned in my second review, that large fields would be required in metallic systems to overcome the problem of nonlinear effects. The authors contradict my statement by citing their result in Ref. 35 where they showed the same mechanism for metallic materials with perpendicular magnetic anisotropy in devices made from CoNi. However, in their publication in Ref. 35 the authors study confined

spin waves in a 500nm disc, which are not propagating and where no information transport could be realized. And still the applied magnetic field was 200mT. Hence, my argument that the manuscript and data are nice and solid, but for a very specialized geometry, still holds and still doesn't allow for CMOS compatibility. If the authors showed the true amplification of propagating spin waves in samples similar as analyzed in Ref.35, then the claim of CMOS compatibility is justified, but still it would be an incremental achievement considering the results in Ref.35 and the present manuscript, because no new physical phenomena has been observed or a known mechanism has been utilized in a novel way.

Reply:

We mentioned Ref. 35 to prove that the shape anisotropy can be compensated by perpendicular magnetic anisotropy in CMOS-compatible materials. Once this compensation is achieved, large fields are not required to magnetize the sample at an arbitrary angle. Therefore, the statement that our approach can also be used with CMOS-compatible materials is completely true. However, since this statement is not central to our work, we decided to remove the sentence mentioning CMOS compatibility in the conclusion section to avoid any misinterpretation.

The Reviewer writes:

Concerning the breakdown field: I calculated the electric field in SI-units (V/m), whereas the authors replied in V/cm, which I did not see in their first rebuttal letter. Hence, the authors are correct, the actual electric field is smaller than the breakdown field they mentioned. However, I believe they are wrong when stating that a breakdown could be avoided by inserting a dc block before the antenna for spin wave excitation. The dc-block protects the signal generator by integrating a capacitor in the signal line. But the ground line is still connected to the ground potential, thus an electric field will build up between antenna and the magnetic material connected to the dc input.

Reply:

We are pleased that the misunderstanding with different breakdown field values has finally been resolved. Regarding the dc decoupling of the ground line, we would like to note that DC-blocks, in which capacitors are integrated in both the signal and the ground lines, are standard microwave components (see, e.g., https://en.wikipedia.org/wiki/DC_block).